# The Effectiveness of Computer-Assisted Cognitive Rehabilitation and the Degree of Recovery in Patients with Traumatic Brain Injury and Stroke

**DOI:** 10.3390/jcm10245728

**Published:** 2021-12-07

**Authors:** Hyunwoo Jung, Jae-Gyeong Jeong, Youn-Soo Cheong, Tae-Woo Nam, Ju-Hyun Kim, Chan-Hee Park, Eunhee Park, Tae-Du Jung

**Affiliations:** 1Department of Rehabilitation Medicine, Kyungpook National University Hospital, Daegu 41944, Korea; hwjung_87@naver.com (H.J.); cloud90524@naver.com (J.-G.J.); yscheong@gmail.com (Y.-S.C.); n0530@daum.net (T.-W.N.); kjoohyun88@gmail.com (J.-H.K.); chany9090@gmail.com (C.-H.P.); 2Department of Rehabilitation Medicine, Kyungpook National University Chilgok Hospital, Daegu 41404, Korea; 3Department of Rehabilitation Medicine, School of Medicine, Kyungpook National University, Daegu 41944, Korea

**Keywords:** traumatic brain injury, stroke, computer-assisted cognitive rehabilitation, computerized neuropsychological test

## Abstract

*Objectives*: To determine the effectiveness of computer-assisted cognitive rehabilitation and compare the patterns of cognitive function recovery occurring in both traumatic brain injury (TBI) and stroke. *Methods:* A total of 62 patients were finally enrolled, consisting of 30 with TBI and 32 with stroke. The patients received 30 sessions of computer-assisted cognitive rehabilitation (Comcog) five times per week. Each session lasted for 30 min. Before and immediately after cognitive rehabilitation, all patients were evaluated by computerized neuropsychological test (CNT), Mini-Mental State Examination (MMSE), and modified Barthel index (MBI). *Results:* We analyzed the differences between pre- and post-cognitive rehabilitation in each TBI and stroke group. Significant differences were observed in MMSE, MBI, and some CNT contents, including digit span forward, verbal learning, verbal learning delayed recall, visual span forward, visual span backward, visual learning, trail making test A and B, and intelligence quotient (IQ) in the TBI group (*p* < 0.05). In the stroke group, in addition to significant differences that appeared in the TBI group, additional significant differences in the digit span backward, visual learning delayed recall, auditory continuous performance test (CPT), visual CPT, and card sorting test. We compared the difference values at pre- and post-cognitive rehabilitation for cognitive recovery between the TBI and stroke groups. All contents, except the digital span forward, visual learning, word-color test, and MMSE, had greater mean values in the stroke group; and thus, statistically significant higher values were observed in the visual span forward and card sorting test (*p* < 0.05). *Conclusion:* Most evaluation results showed improvement and the evaluation between the TBI and stroke groups also showed significant differences in cognitive functions in addition to more CNT contents, which significantly change in the stroke group. The stroke group showed a high difference value in most CNT contents. Therefore, those with stroke in the focal brain region tend to have better cognitive function recovery after a computer-assisted cognitive rehabilitation than those with TBI, which could cause diffuse brain damage and post-injury inflammation.

## 1. Introduction

Acquired brain injury, including traumatic brain injury (TBI) and stroke, causes several neurological deficits, resulting in serious problems for the patients and public health [1,2]. The complications for each individual vary and depend on the site of lesion and severity of the injury. Among other complications, the impaired cognitive function becomes an obstacle that devestates both patients and caregivers and is not only an obstacle to the rehabilitation program [3], it exerts a detrimental effect on the degree of functional status and disability [4]. Therefore, cognitive rehabilitation is necessary for the effective management of acquired brain injuries.

Cognitive rehabilitation was started for patients with brain injuries during World War I for improved survival [5]. Since then, researchers have developed various cognitive rehabilitation techniques and models [6]. In general, two main categories have been established in cognitive rehabilitation techniques, conventional (paper/pencil exercises) and computerized rehabilitation. Both cognitive rehabilitation techniques retrain the patient’s attention and concentration, visual processing, language, memory, reasoning and problem solving, and executive function deficits [7,8,9,10]. Conventional methods are manual exercises with the therapist, whereas computerized rehabilitation uses game-like programs, multimedia and informatics resources with specific hardware and software systems [11,12,13,14,15]. Computer-assisted rehabilitation is an increasingly effective method for patients with acquired brain injury. A recent systematic review shows evidence for computer-assisted rehabilitation in processing and memory ability [16]. Another study showed that computer-assisted rehabilitation is more effective than conventional rehabilitation in digit and visual span, visual learning, and auditory and visual continuous performance [17].

Some commonalities are observed between the TBI and stroke. Since TBI and stroke differ in etiology as well as incidence age and incidence site, a difference in the degree/characteristic/modal/recovery of cognitive impairment is inevitable [18]. First, TBI is characterized by hemorrhagic cerebral contusions originating from acceleration/deceleration with or without striking the head to an object [19]. Diffuse axonal injury with hemorrhagic contusion affects all brain areas in addition to noticeable hemorrhage [20]. The spectrum of cognitive deficits following TBI includes attentional dysfunction, anterograde memory disturbance, language abnormalities, visuospatial deficits, and executive dysfunction [21]. Post-stroke cognitive impairment in both ischemic and hemorrhagic stroke commonly occurs but with diverse clinical manifestations [22]. Deficits are present in various cognitive domains, such as executive function, memory, visuospatial ability, and/or language [23]. However, unlike TBI, instead of receiving injury over the entire brain area, relatively limited problems related to the lesion location increase.

As described above, the pattern and degree of cognitive impairment recovery in patients with stroke and TBI may differ [24]. For this, several studies have been conducted on the recovery of cognitive function in stroke and TBI, respectively [25,26,27]. However, a few studies have addressed the recovery pattern differences between the two diseases. Therefore, this study aimed to determine the effectiveness of computer-assisted rehabilitation and compare the patterns of cognitive function recovery occurring in both TBI and stroke. Thus, the cognitive dysfunction associated with stroke and TBI could be better understood and helpful in predicting the prognosis.

## 2. Materials and Methods

### 2.1. Patients

From December 2013 to October 2019, patients with stroke and TBI who were referred to the rehabilitation department of Chilgok Kyungpook National University Hospital were recruited. Inclusion criteria included (1) patients diagnosed with stroke or TBI using computed tomography and magnetic resonance imaging and (2) patients with impaired cognition (MMSE score of ≤27).

Patients were excluded from the study in the following cases: (1) the presence of a previous central nervous system lesion such as TBI, stroke, brain tumor, and epilepsy; (2) an impossible one-step obey command due to higher brain dysfunction (aphasia or hemispatial neglect) or poor cooperation; (3) the presence of a visual or hearing impairment that interferes with cognitive rehabilitation; and (4) unstable vital signs.

When we reviewed the charts, a total of 854 patients with TBI or stroke were recruited, excluding 757 who met the exclusion criteria in a retrospective study. Of the remaining 97 patients, 35 who did not complete a cognitive rehabilitation program or did not undergo follow-up evaluation and were excluded. As a result, 62 people were finally enrolled, consisting of 30 with TBI and 32 with stroke. The retrospective study was approved by the institutional review board of the hospital (IRB No. 2019-05-008).

### 2.2. Computer-Assisted Cognitive Rehabilitation

Participants underwent 30 sessions of computer-assisted cognitive rehabilitation (Comcog ^®^, Version 1.0, Maxmedica, Seoul, Korea) [28] five times per week. Each session lasted for 30 min per time. The Comcog is a computer-assisted cognitive training system that has been used for years in South Korea [29]. The system provides 10 training activities: 2 auditory processing tasks that assess response time during auditory stimulation; 2 visual processing tasks that assess response time during visual stimulation; 2 selective attention tasks that track attention in distraction; 3 working memory tasks that assess recognition and recall memory using visual, auditory, and multisensory stimulation; and 1 emotional attention task that assesses responses to pleasant or unpleasant stimulation [29].

### 2.3. Assessments

Before starting the cognitive rehabilitation, all patients underwent a computerized neuropsychological test (CNT), a Mini-Mental Status Examination (MMSE), and a modified Barthel index (MBI) performed by occupational therapists. After 30 sessions of Comcog, we reevaluated CNT, MMSE and MBI to determine the cognitive function improvement.

The computerized neuropsychological test (CNT, Version 4.0, Maxmedica, Seoul, Korea) was used for the cognitive function test. The CNT is composed of five components: Verbal memory, visual memory, attention, visuomotor coordination, and high cognition test, which are all computerized as one program (Figure 1). There are tests for each component and the results were presented based on a percentage. Then, the T-score was calculated [30].

#### 2.3.1. Verbal Memory Test

Digit span test: Digit span forward test follows the number played through the computer speaker as it is and a digit span backward test speaks in reverse.Verbal learning test: A total of 15 target words are heard through the computer speaker, recalled in any order and the same target word is repeated five times. After listening to the target word five times, 15 new words are heard and then recalled for blocking and then the 15 target words are recalled again. After 20 min, 15 target words are recalled again and a word list containing 30 words including the target words is presented on the screen and the target words are searched for the delayed recall test.

#### 2.3.2. Visual Memory Test

Visual span test: When nine circles flash in sequence on the screen, a forward visual span test can memorize the order and be clicked with a mouse. A backward visual span test repeats it backward; therefore, the most memorized number is calculated as a score.Visual learning test: After presenting 15 figures on the screen, the patient was instructed to remember the figures. Then, the previously presented figures among 30 figures with the added 15 figures that were not presented were added. After repeating five times in the same way, patients were asked to recall the 15 figures that were previously presented after 20 min for the delayed recall test.

#### 2.3.3. Attention Test

Auditory continuous performance test (CPT): An auditory CPT allows the patient to press a button when the patient hears the number three while playing multiple numbers through the speaker. The response time is scored.Visual CPT: A visual CPT allows the patient to press a button when the patient sees number 3 while showing multiple numbers through the monitor. The response time is scored.Word-color test: It is a test that lists 24 letters of “green,” “blue,” “yellow,” and “red” and the corresponding colors to read them as quickly as possible. Several tests consist of (A) reading black letters, (B) the test to read the color by presenting each color square, (C) the test to read letters that match the color of the letter, and (D) the test to read the color of the letter whose color that composes the letter does not match the letter. The response time is scored.

#### 2.3.4. Visuomotor Coordination Test

Trail making test (TMT): TMT-A is a test that connects numbers up to 25 on the screen, respectively. TMT-B connects the numbers and letters alternately, respectively.

#### 2.3.5. High Cognition Test

Card sorting test: It is a test in which the reference cards with different shapes, colors, and numbers are presented, a problem card and one of three standards are presented and a reference card corresponding to the same standard should be selected.

### 2.4. Statistical Analyses

Data were statistically analyzed using IBM SPSS version 23 (SPSS, Inc., Chicago, IL, USA). The student’s t-test was used to determine significant differences in the CNT contents, MMSE, and MBI between the TBI and stroke groups. To confirm the significant differences in CNT contents, MMSE, and MBI at pre- and post-cognitive rehabilitation in each of the TBI and stroke groups, a paired t-test was used. A value of *p* <  0.05 was considered statistically significant.

## 3. Results

### 3.1. Participant Characteristics

A total of 62 patients (30 TBI and 32 stroke patients) participated in this study. Among stroke patients, 12 had right hemispheric lesion, 18 had left hemispheric lesion and 2 had bilateral lesion. Also, 14 had cerebral infarction and 18 had cerebral hemorrhage and all stroke patients had lesions in the supratentorial area. We investigated the age, sex, duration from disease onset to the initial CNT evaluation (post-onset duration), duration of cognitive rehabilitation by Comcog (cognitive rehabilitation duration), initial MMSE and initial MBI as the descriptive data. These data are summarized in Table 1.

### 3.2. Comparison of Evaluation Pre- and Post-Cognitive Rehabilitation by Comcog

We analyzed the differences between pre- and post-cognitive rehabilitation in the TBI and stroke groups. Significant differences in MMSE, MBI, and some of CNT contents including digit span forward, verbal learning, verbal learning delayed recall, visual span forward, visual span backward, visual learning, TMT-A, TMT-B, and IQ were observed in the TBI group (*p* < 0.05). Moreover, significant differences were observed in MMSE, MBI and all contents of CNT, except for WCT in the stroke group (Table 2). In the stroke group, in addition to significant differences that also appeared in the TBI group, additional significant differences were observed in the digit span backward, visual learning delayed recall, auditory CPT, visual CPT, and card sorting test.

### 3.3. Comparison between TBI and Stroke Groups

We also compared the difference in values at pre- and post-cognitive rehabilitation for cognitive recovery TBI and stroke groups. All contents except the digital span forward, visual learning, word-color test, and MMSE had greater mean values in the stroke group and statistically significant higher values in the visual span forward and card sorting test (*p* < 0.05) (Table 3).

## 4. Discussion

In our study, we attempted to investigate the clinical effectiveness of computer-assisted cognitive rehabilitation in patients with acquired brain injury and the different treatment effects in patients between the TBI and stroke groups. First, comparing the contents of cognitive function evaluation at pre- and post-cognitive rehabilitation, statistically significant improvements were observed in most CNT items. Similar to the results of previous studies, computer-assisted cognitive rehabilitation has been effective in recovering cognitive function in patients with acquired brain injury [16,29,31]. Using the CNT assessment, different abilities of cognitive functions were assessed and each cognitive function subcategory was effective for recovery [30].

First, in the descriptive characteristic of Table 1, there was no item showing a significant difference between the TBI and stroke groups, so we thought that other variables except Comcog rehabilitation would not affect cognitive recovery in this study. In the TBI and stroke groups, significant improvement in cognitive functions was observed between pre- and post-rehabilitation [14,32,33]. In addition, more CNT contents (digit span backward, visual learning delayed recall, auditory CPT, visual CPT and card sorting test) have been statistically significantly improved in the stroke group. In the comparison between TBI and stroke groups on the difference values at pre- and post-cognitive rehabilitation, the stroke group showed a high difference value in most CNT items and statistically significant higher values were observed in visual span forward and the card sorting test. The reason that the cognitive function improved less in the TBI group than in the stroke group may be suggested by inflammation response after TBI. Indeed, the post-injury inflammation, which was mediated by microglia and macrophage, is influenced by existing neurodegenerative pathology [34,35]. Furthermore, several studies reported that there is a biological link between TBI and Alzheimer’s disease (AD) [36,37,38,39,40]. First, population-based studies demonstrated that TBI during adulthood reduces the time to onset of AD [36]. Second, TBI animal model studies showed increased accumulation of amyloid precursor protein (APP), beta-amyloid(Aβ), and pathological tau protein [37,38,39]. Third, accumulation of APP and extracellular deposition of Aβ peptide in senile plaques has been identified in human brain tissue soon after severe TBI [40]. Since post-injury inflammation may be closely associated with neurodegenerative pathology, recovery of cognitive function after TBI may be less effective than that of cognitive rehabilitation after stroke.

Previous results have been obtained for differences in cognitive function impairment or recovery between TBI and stroke [24,41,42]. In our study, those who have strokes in the focal brain region tend to be better for cognitive function recovery after computer-assisted cognitive rehabilitation than TBI patients in CNT subcategories.

However, our study has a few limitations. First, patients were divided into TBI and stroke; however, the location or size of brain injury was not considered. Since which damaged cognitive domain in the brain or degree of brain injury damage is a factor influencing cognitive function recovery, additional consideration of these factors would have provided more assurance in the prognosis for cognitive function recovery in patients with TBI or stroke. Second, cognition is a function that includes complex elements of executive function, memory, visuospatial ability and/or language; therefore, the treatment and evaluation period in this study can be a bit short to see the pattern of cognitive function recovery. If further studies are conducted with more cognitive rehabilitation sessions and the number of cognitive function evaluations, the difference in cognitive function recovery between TBI and stroke groups could be prominent. Third, the amount of data is too small to develop an artificial intelligence (AI)-based tool that can extract the patterns we are looking for. If AI-based tools can be created based on big data, it is thought that it will be possible to more accurately and quickly predict cognitive recovery in patients with TBI and stroke.

## 5. Conclusions

Analyzing the contents of cognitive function evaluation at pre- and post-computer-assisted cognitive rehabilitation in patients with TBI and stroke, most evaluation results showed improvement and the evaluation of patients with TBI and stroke also showed significant changes in cognitive functions in addition to more CNT contents with significant changes in patients with stroke. We also compared the difference in values at pre- and post-cognitive rehabilitation in the TBI and stroke groups. The stroke group showed a high difference value in most evaluation contents. Therefore, those who have strokes in the focal brain region tend to be better for cognitive function recovery after computer-assisted cognitive rehabilitation than patients with TBI, which can cause diffuse brain damage.

## Figures and Tables

**Figure 1 jcm-10-05728-f001:**
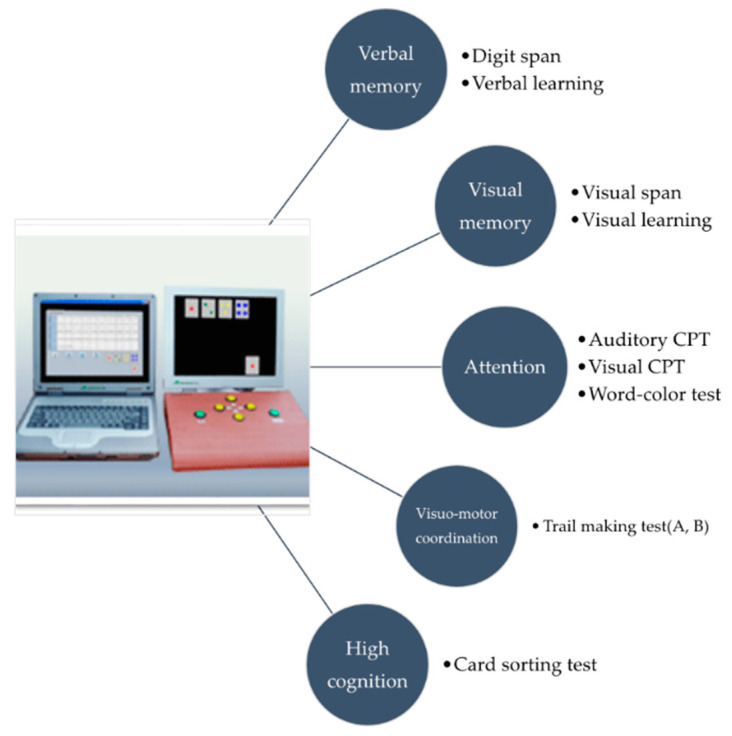
Computerized Neuropsychological Test.

**Table 1 jcm-10-05728-t001:** Descriptive statistics for TBI and stroke groups.

Variables	TBI (*n* = 30)	Stroke (*n* = 32)	*p*-Value ^†^
Gender (*n*)			0.362
Male	22	20	
Female	8	12	
Age (years)	59.03 ± 17.22	57.78 ± 16.66	0.772
Post-onset duration (days)	74.03 ± 43.59	61.13 ± 35.46	0.205
Cognitive rehabilitation duration (days)	52.50 ± 8.46	51.06 ± 6.39	0.452
Initial behavioral parameter			
MMSE	17.70 ± 6.00	20.44 ± 4.99	0.055
MBI	47.00 ± 27.91	42.13 ± 25.71	0.477

Mean ± Standard deviation; TBI, traumatic brain injury; post-onset duration, duration from disease onset to the initial CNT evaluation; cognitive rehabilitation duration, duration of cognitive rehabilitation by Comcog; MMSE, Mini-Mental State Examination; MBI, modified Barthel index; ^†^ *p*-value, comparison between the TBI and stroke groups.

**Table 2 jcm-10-05728-t002:** Analyses of the differences between pre- and post-cognitive rehabilitation in the TBI and stroke groups.

	TBI			Stroke		
	Pre-Rehabilitation	Post-Rehabilitation	*p*-Value ^†^	Pre-Rehabilitation	Post-Rehabilitation	*p*-Value ^‡^
Digital span test_forward	35.33 ± 10.14	40.06 ± 13.98	0.009 *	34.03 ± 9.37	38.03 ± 10.11	0.023 *
Digital span test_backward	32.33 ± 10.01	35.53 ± 11.14	0.135	30.65 ± 6.73	34.65 ± 9.80	0.002 *
Verbal learning test	25.30 ± 6.89	28.30 ± 3.50	0.035 *	27.50 ± 6.10	33.09 ± 10.35	0.001 *
Verbal learning test_delayed	25.20 ± 6.85	28.80 ± 4.95	0.021 *	28.00 ± 7.23	33.65 ± 13.43	0.002 *
Visual span test_forward	30.90 ± 9.70	34.30 ± 9.96	0.017 *	28.87 ± 8.87	37.12 ± 10.43	0.000 *
Visual span test_backward	29.66 ± 7.26	34.40 ± 10.33	0.005 *	28.65 ± 8.58	35.03 ± 9.33	0.000 *
Visual learning test	34.43 ± 11.27	43.13 ± 9.05	0.001 *	36.34 ± 14.70	44.84 ± 10.33	0.001 *
Visual learning test_delayed	39.46 ± 13.89	43.23 ± 13.79	0.160	42.03 ± 16.60	48.75 ± 10.74	0.027 *
Auditory CPT	28.36 ± 9.59	30.03 ± 17.26	0.545	30.06 ± 9.14	33.78 ± 11.23	0.004 *
Visual CPT	33.23 ± 17.14	38.66 ± 20.89	0.152	37.31 ± 17.91	44.75 ± 19.71	0.013 *
Trail making test A	21.86 ± 14.75	27.76 ± 12.68	0.008 *	21.37 ± 15.18	29.75 ± 11.73	0.001 *
Trail making test B	10.40 ± 17.63	17.36 ± 20.43	0.012 *	13.59 ± 16.10	20.96 ± 19.37	0.001 *
Word-color test	27.93 ± 11.81	31.36 ± 10.28	0.103	29.03 ± 7.62	29.37 ± 7.75	0.835
Card sorting test	12.50 ± 21.94	18.16 ± 26.00	0.055	14.53 ± 19.50	29.34 ± 21.53	0.000 *
IQ	38.43 ± 14.80	46.46 ± 18.81	0.008 *	35.87 ± 16.71	46.56 ± 15.14	0.002 *
MMSE	17.70 ± 6.00	22.73 ± 4.66	0.000 *	20.44 ± 4.99	24.06 ± 4.13	0.000 *
MBI	47.00 ± 27.91	65.90 ± 26.84	0.000 *	42.13 ± 25.71	65.59 ± 23.99	0.000 *

Mean ± Standard deviation; CPT, continuous performance test; MMSE, Mini-Mental State Examination; MBI, modified Barthel index; ^†^ *p*-value, comparison between pre- and post-cognitive rehabilitation in the TBI group; ^‡^ *p*-value, comparison between pre- and post-cognitive rehabilitation in the stroke group; * *p*-value < 0.05.

**Table 3 jcm-10-05728-t003:** Comparison between the TBI and stroke groups on difference values at pre- and post-cognitive rehabilitation.

	TBI	Stroke	*p*-Value ^†^
Digital span test_forward	4.73 ± 9.32	4 ± 9.49	0.76
Digital span test_backward	3.2 ± 11.39	4 ± 6.64	0.735
Verbal learning test	3 ± 7.43	5.59 ± 8.37	0.203
Verbal learning test_delayed	3.6 ± 8.05	5.66 ± 9.66	0.368
Visual span test_forward	3.4 ± 7.32	8.25 ± 8.71	0.021 *
Visual span test_backward	4.73 ± 8.43	6.38 ± 8.48	0.448
Visual learning test	8.7 ± 12.37	8.5 ± 12.78	0.95
Visual learning test_delayed	3.77 ± 14.30	6.72 ± 16.35	0.454
Auditory CPT	1.67 ± 14.88	3.72 ± 6.77	0.483
Visual CPT	5.43 ± 20.24	7.44 ± 16.01	0.666
Trail making test A	5.9 ± 11.44	8.38 ± 12.21	0.414
Trail making test B	6.97 ± 14.14	7.38 ± 10.80	0.898
Word-color test	3.43 ± 11.17	0.34 ± 9.26	0.24
Card sorting test	5.47 ± 15.50	14.81 ± 20.16	0.049 *
IQ	8.03 ± 15.55	10.69 ± 17.47	0.531
MMSE	5.03 ± 4.15	3.63 ± 2.42	0.113
MBI	18.9 ± 14.94	23.47 ± 16.53	0.259

Mean ± Standard deviation; CPT, continuous performance test; MMSE, Mini-Mental State Examination; MBI, modified Barthel index; ^†^ *p*-value, comparison between the TBI and stroke groups on difference in values at pre- and post-cognitive rehabilitation; * *p*-value < 0.05.

## Data Availability

Available upon reasonable request.

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
