# Peer review of "The Effectiveness of Computer-Assisted Cognitive Rehabilitation and the Degree of Recovery in Patients with Traumatic Brain Injury and Stroke"

_jcm, 2021, doi:10.3390/jcm10245728_

Round 1
Reviewer 1 Report
This paper examines whether there is a difference in the impact of computer-assisted rehabilitation (Comcog) on cognitive recovery between TBI and stroke cases.
The results are interesting in that they conclude that stroke patients had more items that showed improvement with Comcog.
However, I think that the following points need to be described in more detail.
1) The breakdown of stroke cases should include the number of left and right hemisphere injuries, the number of cerebral infarctions and cerebral hemorrhages and, if possible, the location of the injury.
2) In addition, please provide a discussion of the reasons for any improvement over TBI.
3) In addition, the presence or absence of higher brain dysfunction (e.g., aphasia, spatial neglect) in TBI and stroke cases should be indicated, and the results should be interpreted accordingly.
Author Response
Response to Reviewers
We thank editor for their constructive and valuable comments. We have addressed their specific concerns below and have made the suggested changes, which were helpful for improving the manuscript. Main changes in the manuscript are highlighted in blue letters.
Reviewer 1 comments
- The breakdown of stroke cases should include the number of left and right hemisphere injuries, the number of cerebral infarctions and cerebral hemorrhages and, if possible, the location of the injury.
Response: Thank you for your valuable comment. Among stroke patients, 12 patients had right hemispheric lesion, 18 patients had left hemispheric lesion and 2 patients had bilateral lesion. Also, 14 had cerebral infarction and 18 had cerebral hemorrhage and all stroke patients had lesions in the supratentorial area. These contents were added to 3.1 participants characteristics of the Result section as follow:
“Among stroke patients, 12 had right hemispheric lesion, 18 had left hemispheric lesion and 2 had bilateral lesion. Also, 14 had cerebral infarction and 18 had cerebral hemorrhage and all stroke patients had lesions in the supratentorial area.”
- In addition, please provide a discussion of the reasons for any improvement over TBI.
Response: Thank you for your valuable comment. We described why recovery of cognitive function after TBI may be less effective than that of cognitive rehabilitation after stroke in the Discussion section as follows:
“The reason that the cognitive function improved less in the TBI group than in the stroke group may be suggested by inflammation response after TBI. Indeed, the post-injury inflammation, which was mediated by microglia and macrophage, is influenced by existing neurodegenerative pathology [34,40]. Furthermore, several studies reported that there is a biological link between TBI and Alzheimer’s disease (AD) [35-39]. First, population-based studies demonstrated that TBI during adulthood reduces the time to onset of AD [35]. Second, TBI animal model studies showed increased accumulation of amyloid precursor protein (APP), beta-amyloid(Aβ), and pathological tau protein [36-38]. Third, accumulation of APP and extracellular deposition of Aβ peptide in senile plaques has been identified in human brain tissue soon after severe TBI [39]. Since post-injury inflammation may be closely associated with neurodegenerative pathology, recovery of cognitive function after TBI may be less effective than that of cognitive rehabilitation after stroke.”
- Kokiko-Cochran, O.N. and J.P. Godbout, The inflammatory continuum of traumatic brain injury and Alzheimer’s disease. Frontiers in immunology, 2018. 9: p. 672.
- Nemetz, P.N., et al., Traumatic brain injury and time to onset of Alzheimer's disease: a population-based study. American journal of epidemiology, 1999. 149(1): p. 32-40.
- Webster, S.J., et al., Closed head injury in an age-related Alzheimer mouse model leads to an altered neuroinflammatory response and persistent cognitive impairment. Journal of Neuroscience, 2015. 35(16): p. 6554-6569.
- Washington, P.M., et al., Experimental traumatic brain injury induces rapid aggregation and oligomerization of amyloid-beta in an Alzheimer's disease mouse model. Journal of neurotrauma, 2014. 31(1): p. 125-134.
- Itoh, T., et al., Expression of amyloid precursor protein after rat traumatic brain injury. Neurological research, 2009. 31(1): p. 103-109.
- Roberts, G., et al., Beta amyloid protein deposition in the brain after severe head injury: implications for the pathogenesis of Alzheimer's disease. Journal of Neurology, Neurosurgery & Psychiatry, 1994. 57(4): p. 419-425.
- Johnson, V.E., et al., Inflammation and white matter degeneration persist for years after a single traumatic brain injury. Brain, 2013. 136(1): p. 28-42.
- In addition, the presence or absence of higher brain dysfunction (e.g., aphasia, spatial neglect) in TBI and stroke cases should be indicated, and the results should be interpreted accordingly.
Response: Thank you for your valuable comment. We excluded patients with higher brain dysfunction (aphasia, spatial neglect) that made it difficult to receive Comcog rehabilitation. It is indicated in 2. Materials in the Methods section of the manuscript.
Reviewer 2 Report
The paper presents the effectiveness of computer assisted rehabilitation (Comcog) and compares the patterns of cognitive function recovery for post-stroke and traumatic brain injury patients.
It seems that the focus is set on the methodology of assessment and not on the rehabilitation itself. It would have been interesting to see a strong relation between the rehabilitation and the results measuring technique, thus increasing the paper readability.
There are a few unclear issues regarding the results and/or the evaluation methodology.
First, can you detail how did you obtain the indicators in table 2?
Regarding the patient’s data in Table 1, did you consider their age? How did this influence the results?
Is there a way to assess the cognitive degree of impairment for these patients? Or did you consider all of them at the same level?
How would the training beginning time (in days after the event) influence the results?
Did you consider developing an Artificial Intelligence based tool to help extract the patterns you are looking for? Of course, a large set of data would be required.
Author Response
Response to Reviewers
We thank editor for their constructive and valuable comments. We have addressed their specific concerns below and have made the suggested changes, which were helpful for improving the manuscript. Main changes in the manuscript are highlighted in blue letters.
Reviewer 2 comments
- First, can you detail how did you obtain the indicators in table 2?
Response: Thank you for your valuable comment. We evaluated CNT scores before and immediately after Comcog rehabilitation. We added more detail in computer-assisted cognitive rehabilitation in the revised manuscript as follows:
“2.2 Computer-assisted Cognitive Rehabilitation
Participants underwent 30 sessions of computer-assisted cognitive rehabilitation (Comcog®, Maxmedica) [29] five times per week. Each session lasted for 30 min per time. The Comcog is a computer-assisted cognitive training system that has been used for years in South Korea [30]. The system provides 10 training activities: 2 auditory processing tasks that assess response time during auditory stimulation; 2 visual processing tasks that assess response time during visual stimulation; 2 selective attention tasks that track attention in distraction; 3 working memory tasks that assess recognition and recall memory using visual, auditory, and multisensory stimulation; and 1 emotional attention task that assesses responses to pleasant or unpleasant stimulation [30].”
- Kim, Y.H., et al., Development of computer-assisted memory rehabilitation programs for the treatment of memory dysfunction in patients with brain injury. Journal of Korean Academy of Rehabilitation Medicine, 2003. 27(5): p. 667-674.
- Kim, Y.H., et al., Effect of computer-assisted cognitive rehabilitation program for attention training in brain injury. Journal of Korean Academy of Rehabilitation Medicine, 2003. 27(6): p. 830-839.
- Regarding the patient’s data in Table 1, did you consider their age? How did this influence the results?
Response: Thank you for your valuable comment. There was no statistically significant difference in age between the TBI and stroke groups (p=0.772), so the effect of age on cognitive function recovery was not considered. So, we added it in the revised manuscript as follow:
“First, in the descriptive characteristic of Table 1, there was no item showing a significant difference between the TBI and stroke groups, so we thought that other variables except Comcog rehabilitation would not affect cognitive recovery in this study.”
- Is there a way to assess the cognitive degree of impairment for these patients? Or did you consider all of them at the same level?
Response: Thank you for your valuable comment. The overall cognitive impairment was assessed through the MMSE score to determine whether the person had the cognitive ability to perform CNT and Comcog. The MMSE scores are presented in Table 1. And we assessed the cognitive degree of impairment with various items of CNT. In other words, cognitive impairment was assessed by MMSE and CNT.
- How would the training beginning time (in days after the event) influence the results?
Response: Thank you for your valuable comment. Training start time means the time from disease onset to initial CNT evaluation. This means the time from disease onset to the start of Comcog rehabilitation, which can act as a variable affecting cognitive function recovery if there is a statistically significant difference between the start time of Comcog rehabilitation in the stroke and TBI groups. But there was no statistically significant difference in age between the TBI and stroke groups (p=0.205). So, we added this in our discussion as follow:
“First, in the descriptive characteristic of Table 1, there was no item showing a significant difference (p-value < 0.05) between the TBI and stroke groups, so we thought that other variables except Comcog rehabilitation would not affect cognitive recovery in this study.”
- Did you consider developing an Artificial Intelligence based tool to help extract the patterns you are looking for? Of course, a large set of data would be required.
Response: Thank you for your valuable comment. The amount of data is too small to develop an AI-based tool that can extract the patterns we are looking for, so it was not considered in this study. We added this limitation in the Discussion section as follow:
“Third, the amount of data is too small to develop an artificial intelligence (AI)-based tool that can extract the patterns we are looking for. If AI tools can be created based on big data, it is thought that it will be possible to more accurately and quickly predict cognitive recovery in patients with TBI and stroke.”
Round 2
Reviewer 1 Report
I think this has been corrected in response to my comment
Reviewer 2 Report
The authors have addressed all my comments successfully.